# ERK1/2 Activity Is Critical for the Outcome of Ischemic Stroke

**DOI:** 10.3390/ijms23020706

**Published:** 2022-01-09

**Authors:** Constanze Schanbacher, Michael Bieber, Yvonne Reinders, Deya Cherpokova, Christina Teichert, Bernhard Nieswandt, Albert Sickmann, Christoph Kleinschnitz, Friederike Langhauser, Kristina Lorenz

**Affiliations:** 1Institute of Pharmacology and Toxicology, University of Würzburg, 97078 Würzburg, Germany; constanze.schanbacher@uni-wuerzburg.de; 2Leibniz-Institut für Analytische Wissenschaften-ISAS-e.V., 44139 Dortmund, Germany; yvonne.reinders@isas.de (Y.R.); ch.mathejka@gmx.de (C.T.); albert.sickmann@isas.de (A.S.); 3Department of Neurology, University Hospital Würzburg, 97080 Würzburg, Germany; Bieber_M@ukw.de; 4Institute of Experimental Biomedicine I, University Hospital Würzburg, 97080 Würzburg, Germany; deya.cherpokova@gmail.com (D.C.); bernhard.nieswandt@virchow.uni-wuerzburg.de (B.N.); 5Rudolf Virchow Center, University of Würzburg, 97080 Würzburg, Germany; 6Department of Neurology and Center for Translational Neuro- and Behavioral Sciences (C-TNBS), University Hospital Essen, 45147 Essen, Germany; christoph.kleinschnitz@uk-essen.de

**Keywords:** ERK1/2, tMCAO, ischemic stroke, RKIP

## Abstract

Ischemic disorders are the leading cause of death worldwide. The extracellular signal-regulated kinases 1 and 2 (ERK1/2) are thought to affect the outcome of ischemic stroke. However, it is under debate whether activation or inhibition of ERK1/2 is beneficial. In this study, we report that the ubiquitous overexpression of wild-type ERK2 in mice (ERK2^wt^) is detrimental after transient occlusion of the middle cerebral artery (tMCAO), as it led to a massive increase in infarct volume and neurological deficits by increasing blood–brain barrier (BBB) leakiness, inflammation, and the number of apoptotic neurons. To compare ERK1/2 activation and inhibition side-by-side, we also used mice with ubiquitous overexpression of the Raf-kinase inhibitor protein (RKIP^wt^) and its phosphorylation-deficient mutant RKIP^S153A^, known inhibitors of the ERK1/2 signaling cascade. RKIP^wt^ and RKIP^S153A^ attenuated ischemia-induced damages, in particular via anti-inflammatory signaling. Taken together, our data suggest that stimulation of the Raf/MEK/ERK1/2-cascade is severely detrimental and its inhibition is rather protective. Thus, a tight control of the ERK1/2 signaling is essential for the outcome in response to ischemic stroke.

## 1. Introduction

Cardiovascular diseases are the leading cause of mortality worldwide, especially ischemic disorders such as stroke, which are responsible for substantial mortality and morbidity rates [1]. Currently, there are hardly any treatment options for ischemic stroke and the existing options are limited to the acute setting after the ischemic insult [2,3,4]. Therefore, it is of therapeutic interest to understand the mechanisms involved in brain damage due to ischemic stroke and to identify intervention strategies.

Protein kinases that are of central scientific interest in ischemia reperfusion (I/R) injury are the extracellular signal-regulated kinases 1 and 2 (ERK1/2) [5,6]. ERK1/2 are ubiquitously expressed throughout the body and are part of the Raf/mitogen-activated protein kinase (MEK)/ERK1/2 signaling cascade [7]. Activation of this cascade is involved in proliferation, differentiation, cell death, and inflammation [5,6,8]. The impact of ERK1/2 signaling in stroke was analyzed using different MEK/ERK1/2 inhibitors [9,10] or by transient overexpression of the endogenous Raf-kinase inhibitor protein (RKIP) [11,12], or agents that stimulate the Raf/MEK/ERK1/2-cascade such as β-hydroxybutyrate [13] or microribonucleic acid-1 (miR-1) [14]. Several of these studies suggest that ERK1/2 act as neuroprotectants by suppressing oxidative stress, mitochondrial-dependent apoptosis [13], and neuronal apoptosis [14]. In contrast, there are also studies that suggest that ERK1/2 inhibition protects from inflammation, apoptosis, and blood–brain barrier (BBB) damage [15,16,17,18]. Thus, it is still under debate whether the activation or inhibition of this signaling pathway is protective after stroke [6].

In this study, we aimed to investigate the role of ERK1/2 after acute cerebral ischemia in transgenic mice using constitutive overexpression of proteins that impact on ERK activation, i.e., mice that ubiquitously overexpress ERK2^wt^, RKIP^wt^, or RKIP^S153A^ under the control of the “CAG” promoter (cytomegalovirus enhancer, β-actin intron, and bovine globin poly-adenylation signal promoter) [19,20]. These mouse models allow an alternative approach to untangle the previous contradictory findings since none of the previous studies have compared ERK1/2 activating and inhibitory effects side-by-side. For the generation of these mice, identical constructs and genetic mouse background were applied. Interestingly, our study shows rather opposing effects of ERK2^wt^ and the endogenous Raf/MEK/ERK1/2-inhibitory protein RKIP on the outcome after stroke.

## 2. Results

### 2.1. ERK2^wt^ Overexpression Leads to Massive Damage after Stroke

To assess the impact of ERK1/2 signaling in ischemic stroke, we generated mice with a ubiquitous overexpression of ERK2^wt^, RKIP^wt^, and RKIP^S153A^ under the control of the CAG promoter (Appendix A). RKIP is an endogenous inhibitor of the upstream kinase of ERK1/2, i.e., Raf-1. It controls in particular two signaling pathways: G protein coupled receptors and Raf/MEK/ERK1/2 signaling dependent on its phosphorylation status on serine 153 [21]. The phosphorylation of this site by protein kinase C was described to release RKIP from Raf binding [19]. In line with this, RKIP^S153A^, a mutant of RKIP that cannot be phosphorylated at serine 153, is primarily known as a Raf inhibitor [19,20].

The transgenic (tg) mice and their wild-type (wt) littermates were subjected to 45 min of transient middle cerebral artery occlusion (tMCAO) to assess the potential detrimental or protective effects of ERK1/2 signaling. The histological assessment of infarct size by 2,3,5-triphenyltetrazolium chloride (TTC) staining 24 h after the induction of stroke revealed that an overexpression of ERK2^wt^ led to an increase in infarct size (Figure 1A). In line with this, neurological deficits were also increased compared to wt mice as assessed by the Bederson score and the grip test, which evaluate motoric and coordinative deficits (Figure 1B). In contrast, RKIP^wt^ and RKIP^S153A^ expression significantly reduced infarct volume (Figure 1A) and improved neurological functions (Figure 1B) compared to wt controls. Of note, RKIP^wt^ and RKIP^S153A^ mice were indistinguishable in these analyses, suggesting that the RKIP effects are most likely due to its binding to Raf.

Similar data were obtained three days after I/R in ERK2^wt^ and RKIP overexpressing mice (Figure 2). These results give the first hint that ERK1/2 signaling may be rather detrimental in ischemic injury.

### 2.2. Overexpression of ERK2^wt^ Decreases Blood–Brain Barrier Stability

Disruption of the BBB is one of the consequences in response to cerebral ischemia. Endothelial cells are damaged and secrete the vasoconstrictive endothelin-1 that is associated with a poor prognosis after stroke [22]. The instability of the BBB subsequently leads to the formation of edema that cause further deterioration [23,24]. We therefore evaluated the effects of ERK2^wt^, RKIP^wt^, and RKIP^S153A^ overexpression on BBB function to see if it is responsible for the observed functional outcome/phenotype. Leakage of the BBB was determined by measuring the extravasation of the vascular tracer Evans Blue into the brain parenchyma. After stroke, ERK2^wt^ mice showed a significant increase in Evans Blue extravasation into the ipsilateral hemisphere compared to wt mice (Figure 3A). Of note, the leakage of the BBB in ERK2^wt^ mice even affected the contralateral hemisphere. The stability of the BBB in RKIP^wt^ and RKIP^S153A^ mice was comparable to the wt controls. In line with these findings, mRNA expression levels of endothelin-1 were upregulated in ERK2^wt^ compared to wt and RKIP-tg mice (Figure 3B). Thus, disruption of the BBB seems to contribute to the detrimental phenotype of ERK2^wt^ overexpression. However, the protective effects of RKIP on the infarct size and neurological functions cannot be explained by a prevention of the BBB leakage.

### 2.3. Overexpression of ERK2^wt^ Increases Inflammation after Stroke

Besides leakage, adhesion proteins such as intercellular adhesion molecule 1 (*ICAM1*) or vascular adhesion molecule 1 (*VCAM1*) also change within the BBB after stroke. These proteins are responsible for the anchorage and infiltration of inflammatory cells and contribute to the deterioration after stroke [25,26,27,28]. Indeed, mRNA expression levels of *ICAM1* and *VCAM1* were significantly elevated in ERK2^wt^ mice in comparison to non-transgenic littermates, while the expression levels of these genes were indistinguishable from wt levels in RKIP mice (Figure 4A,B).

To assess whether the increase in the expression of adhesion molecules results in the invasion of inflammatory cells and accounts for the detrimental effects of ERK2^wt^ overexpression, we first measured the gene expression profile of the pro-inflammatory cytokines interleukin-1β (*IL-1β*), interleukin-6 (*IL-6*), and tumor necrosis factor-α (*TNF-α*) [25,26,29,30,31], which contribute to I/R injury [25,26]. In line with the reduced stability of the BBB, mRNA concentrations of all tested cytokines were increased in ERK2^wt^ mice compared to all other genotypes, suggesting the invasion of inflammatory cells. Both RKIP mutants showed a slight but non-significant reduction in the expression levels of the above-mentioned cytokines (Figure 4C–E). To further analyze the impact of ERK2 signaling on inflammation, we determined the number of infiltrated neutrophils in the ipsilateral hemisphere by immunostaining and found a significant increase in the ERK2^wt^ animals and significant decrease in RKIP^wt^ and RKIP^S153A^ mice compared to wt mice (Figure 4F). Thus, ERK2^wt^ overexpression triggers the expression of adhesion molecules, immune cell infiltration, and the secretion of pro-inflammatory cytokines after stroke, while RKIP rather reduces the inflammatory processes.

### 2.4. ERK2^wt^ Overexpression Effects Neuronal Apoptosis

The cytokines secreted by invading immune cells can stimulate neuronal cell death and promote tissue damage after brain ischemia [25,32]. Immunostaining of brain sections with NeuN revealed a slight reduction of neuronal cells in ERK2^wt^ mice in the contralateral as well as in the ipsilateral hemisphere compared to wt mice, whereas RKIP mutants showed a slightly higher number of neurons (Figure 5A,B). Of note, the infarct in the ERK2^wt^ mice extended beyond the midline into the contralateral hemisphere, which validates the massive brain damage in these mice. In line with this, we also found an increased number of apoptotic neurons in the contralateral hemisphere as shown by TUNEL co-staining (Figure 5C). Of note, the number of apoptotic cells in the ipsilateral area were similar in all genotypes including RKIP^wt^ and RKIP^S153A^ (Figure 5D). Altogether, neuronal death and apoptosis contribute—in addition to BBB leakage and inflammation—to the deleterious effect of ERK2^wt^ after I/R.

### 2.5. Proteomic Analyses Revealed Massive Changes in ERK2^wt^ after tMCAO

Thus far, we analyzed mechanisms that are typically associated with ischemic damage after stroke, such as BBB leakage, inflammation, and neuronal death, and found that all of these are involved in the phenotype of ERK2^wt^ mice after tMCAO, while RKIP overexpression impacts inflammation in particular. To assess whether other signaling mechanisms are involved in the development of the observed phenotypes—by RKIP or ERK2^wt^ overexpression, respectively—we performed an untargeted mass-spectrometric analysis of the basal ganglia after tMCAO. Overall, we detected 3878 proteins. First of all, we compared the contralateral basal ganglia (bgc) to the ipsilateral basal ganglia (bgi) of wt mice. These analyses revealed that 214 proteins were significantly regulated in the bgc versus the bgi of wt mice (Appendix A). The comparison of the bgc of wt mice to the RKIP-tg mice revealed a similar number of regulated proteins, i.e., 235 proteins vs. RKIP^wt^ bgi (Appendix A) and 179 vs. RKIP^S153A^ bgi (Appendix A). Interestingly, 365 proteins were regulated between the wt bgc and ERK2^wt^ bgi, indicating a significantly more pronounced difference (Appendix A). These results are in line with the strength of the ERK2^wt^ phenotype and involved mechanisms.

To analyze the tMCAO-mediated differences throughout the genotypes, we then compared wt bgi with ERK2^wt^ bgi and all other genotypes. Of note, most of the proteins with significant changes in ERK2^wt^ bgi vs. wt bgi were similar between wt and RKIP-tg bgi (Figure 6 and Appendix A).

To evaluate the function of the regulated proteins in the ipsilateral hemisphere and their impact on stroke outcome, we assigned the significantly regulated proteins to gene ontology functional categories (GO-term: biological process) using a cut-off for a 2-fold upregulation and for a 0.5-fold downregulation compared to wt bgi. These results show that the regulated proteins were found within the signaling processes commonly involved after ischemic brain injury, such as cell proliferation, cell differentiation, cell growth, cell death, translation, transcription, synaptic plasticity, signal transduction, metabolism, protein transport, vesicle transport, proteolysis, cell adhesion, membrane potential, immune response/inflammation, cytoskeleton organization, and regulation of hydrogen peroxide (Appendix A) [33,34,35]. As these processes were affected in all genotypes, the analyses did not allow the identification of a main mechanism for the respective genotypes. To have a closer look at the regulated proteins that are differentially regulated between the genotypes, we focused on proteins that were either significantly regulated in (i) ERK2^wt^ vs. wt but not in RKIP-tg mice vs. wt or (ii) in RKIP-tg vs. wt but not in ERK2^wt^ vs. wt. Of note is that these analyses revealed a downregulation of neuroprotective proteins in ERK2^wt^ bgi compared to wt bgi, which is in line with the detrimental neurological phenotype of ERK2^wt^ mice. These proteins include Ras-specific guanine nucleotide-releasing factor 1 (Ras-GRF1) and neuroserpin, which were shown to be protective after cerebral ischemia [36,37,38]. In RKIP^wt^ and RKIP^S153A^, a member of the eukaryotic translation initiation factors, i.e., EIF1A, was upregulated compared to that in wt. This family is thought to play a protective role in ischemic stress [39].

## 3. Discussion

In the present study, we demonstrate that the ubiquitous overexpression of ERK2^wt^ is detrimental in a model of ischemic stroke as it increases infarct size and neurological deficits after tMCAO by increasing disruption of the BBB, the immune response, and a loss of neurons. To compare ERK1/2-activating as well as ERK1/2-inhibiting effects within one study and under comparable conditions, mice with an overexpression of a known endogenous ERK1/2 inhibitor, i.e., RKIP^wt^ or RKIP^S153A^, were studied. Interestingly, infarct size and neurological deficits were significantly attenuated in RKIP-tg mice, supporting the hypothesis that ERK1/2 signaling contributes to the detrimental tMCAO phenotype. RKIP overexpression in particular interfered with the tMCAO-induced inflammatory signaling. Thus far, the effects of ERK1/2 in I/R injury have only been approached indirectly with ERK1/2 phosphorylation as a read-out by a direct or indirect modulation of the cascade [9,10,40]. In our study, we show that an increase in ERK2^wt^ expression and thus a certain increase in baseline ERK1/2 activity is detrimental for the outcome of cerebral I/R injury: compared to wt mice, ERK2^wt^ mice displayed a greatly increased infarct size and impaired neurological function (Figure 1). These detrimental effects were associated with an instability in the BBB (Figure 3); an increase in inflammatory cytokines such as *IL-1β*, *IL-6*, and *TNF-α* (Figure 4); and apoptosis (Figure 5). These data are in accordance with other studies that observed that ERK1/2 inhibition by compounds via direct or indirect targeting is protective [15,16,17,18]. In contrast to the studies describing protective effects of ERK1/2 inhibition, there are also reports that classify ERK1/2 as neuroprotectants [40,41]. In their review about the dual roles of ERK1/2 after stroke, Sawe et al. suggest that the stimulus for ERK1/2 activation in brain ischemia is decisive for its protective or detrimental function. ERK1/2 activation due to oxidative stress or inflammatory cytokines induced by the stroke itself or other damaging insults may result in the deterioration of ischemic injury. In contrast, ERK1/2 stimulation by growth factors or other protective compounds as well as preconditioning may prevent ischemic damage through different pathways as they also prevent the release of ERK1/2 triggers such as reactive oxygen species or pro-inflammatory cytokines that are known to induce adverse outcomes in stroke [6]. Our study strongly supports the hypothesis that the activation of ERK1/2 in ischemic stroke can cause massive damage, as shown, for example, by the huge infarct size that extends even to the contralateral hemisphere and by the significantly impaired neurological function after tMCAO. Although, we cannot rule out by our experiments that transient and moderate activation of ERK1/2 or the mode of ERK1/2 activation [42] may be beneficial, our study points in the direction that any induction of even a moderate ERK1/2 activation may constitute a risk for patients.

RKIP is one of the proteins that was previously characterized as a Raf/MEK/ERK1/2-inhibitor and was shown to be protective after stroke. Su et al. found that lentivirus encoding RKIP in the hippocampus and cortex of rat brains protected from cerebral injury by tMCAO [43]. Also, Jung et al. showed that RKIP fused to a cell penetrating BBB-crossing peptide, PEP-1-RKIP, attenuated ischemia-induced ERK1/2 activation and brain damage [44]. In contrast to the reported protective RKIP effects in stroke, Wang et al. found that the knockdown of RKIP is protective and the overexpression of RKIP is rather detrimental for the outcome after tMCAO. Of note, the overexpression of the phosphorylation-deficient RKIP^S153A^ mutant seemed to be rather protective in the setting of their study. The authors concluded that phosphorylated RKIP is responsible for the detrimental outcome after neuronal ischemia and suggested a RKIP-mediated increase in phosphatidylcholine-phospholipase C activity and ERK1/2 activation as a trigger for inflammation and apoptosis [12]. The controversy of the different studies with regard to the detrimental or protective effects of wild-type RKIP may be explained by slightly different experimental settings or RKIP expression levels resulting in potentially different phosphorylation statuses. Taken together, these studies agree that RKIP mediates its protective effects largely as an ERK1/2 inhibitor and thereby support that this is the mechanism responsible for the beneficial RKIP effects.

Our study analyzing ERK1/2 and RKIP functions by side-by-side comparison showed that RKIP overexpression does not target all the detrimental processes triggered by ERK2 activation via ERK2^wt^ expression after tMCAO, as RKIP only affected inflammation but did not influence BBB instability and apoptosis. Thus, ERK1/2 and RKIP may involve additional signaling pathways. Our proteomic analyses revealed some interesting signaling components that may help to understand the detrimental or protective effects of ERK1/2 and RKIP, respectively. Among them is the tissue-type plasminogen activator inhibitor neuroserpin that was downregulated in ERK2^wt^. Neuroserpin was shown to be protective in cerebral ischemia by reducing apoptosis, the proteolytic degradation of basement membrane, and the inflammatory response in microglia [38,45]. For instance, it was shown that the administration of neuroserpin to microglia cells could inhibit the release of *IL-1β* after oxygen glucose deprivation and reperfusion [45]. Thus, the reduced expression levels of neuroserpin in ERK2^wt^ mice might be a reason for the higher *IL-1β* expression compared to RKIP-tg mice as detected by real-time PCR (Figure 4). Therefore, the reduction of this endogenous neuroprotectant in ERK2^wt^ overexpression might account for the observed detrimental phenotype of ERK2^wt^ and may be involved in the increased instability of the BBB, inflammation, and neuronal death. Another interesting candidate is Ras-GRF1, which was also significantly downregulated in ERK2^wt^. Ras-GRF1 together with Ras-GRF2 mediates ERK/CREB activation via the N-methyl-D-aspartate glutamate receptor in neurons and was shown to be protective in stroke as Ras-GRF1/Ras-GRF2 double knockout mice had larger infarct volumes compared to wild-type mice after the induction of cerebral ischemia [36]. Thus, a reduced Ras-GRF1 expression may contribute to the deleterious outcome of ERK2^wt^ overexpression. In RKIP-tg mice, 60kDa SS-A/Ro ribonucleoprotein and ADP ribosylation factor like protein 3, i.e., proteins involved in the smoothened (SMO) signaling pathway, were found to be downregulated. Since SMO activation inhibits the activity of the glutamate transporter 1 and thereby increases extracellular glutamate levels, which have shown to be toxic after ischemic brain damage [46], the observed downregulation in our model may contribute to the protective effects of RKIP after tMCAO.

In summary, our findings show that a ubiquitous overexpression of ERK2^wt^ is severely detrimental in ischemic stroke, whereas an overexpression of the endogenous ERK1/2 inhibitors RKIP and its phosphorylation-deficient mutant RKIP^S153A^ improve stroke outcome for at least up to three days after I/R. Therefore, our data indicate that stimulation of the Raf/MEK/ERK1/2-cascade is harmful and its inhibition is rather protective. Further studies are needed to investigate the exact mechanism and cell types that contribute to these conclusions.

## 4. Materials and Methods

### 4.1. Mice

Transgenic mice with a ubiquitous overexpression of ERK2^wt^ (ERK2^wt^), RKIP^wt^ (RKIP^wt^), or a phosphorylation-deficient mutant of RKIP (RKIP^S153A^) under the control of the vector containing chicken β-actin promoter, cytomegalovirus enhancer, β-actin intron, and bovine globin poly-adenylation signal (CAG) promoter were generated by injection into the pronucleus of fertilized oocytes from FVB/N mice [20,47]. Animal experiments were approved by the Committee on Animal Research of the regional government (Regierung von Unterfranken) and were performed conforming to the US National Institutes of Health Guide for the Care and Use of Laboratory Animals (55.2-2531.01 -46/09 and -72/14). The experiments were designed, conducted, and reported according to the “Animal Research: Reporting of In Vivo Experiments guidelines” [48]. In all experiments, male FVB/N mice aged 8–12 weeks purchased from Charles River (Sulzfeld, Germany) were used and age- and gender-matched littermates were picked as controls. We employed only male mice to minimize fluctuations in our outcome parameters caused by sex-differences and to thereby reduce the number of needed animals as effects of sex-differences have been shown on infarct size and immune response [49]. Further studies including female, comorbid, or older mice are needed to fully understand the effect of ERK1/2 activation or inhibition after stroke. Surgical procedures and the analysis of all parameters were done in a blinded manner for all experiments. The mice were supplied with standard chow diet and water ad libitum. For mouse husbandry, sterilized plastic cages were used and specific pathogen-free conditions were maintained; the housing conditions were as follows: 22 ± 2 °C, 12/12 light/dark cycle, 55 ± 10% humidity, and <400 lux. Hygiene monitoring was conducted on a regular basis four times a year [47].

A priori sample size analysis was done using G*Power software 3.1 (University of Kiel, Germany) to calculate the needed animal number. We assumed that a reduction of infarct size of 25% would be of biological relevance and assumed based on previous studies a standard deviation of 20% of the respective mean values (Cohen’s d = 0.675). Thus, a group size of 10 animals was calculated to be needed to achieve the statistical power (1—type II error) of 0.8 and type I error of < 0.05.

### 4.2. Ischemia Model

Focal cerebral ischemia was induced by 45 min of transient middle cerebral artery occlusion (tMCAO) as described previously [50]. As anesthesia, 2% isoflurane (CP-Pharma, Burgdorf, Germany) in O_2_ was used and body temperature was maintained at 37 °C using a heating plate. After a midline skin incision in the neck, the preparation of the right neck vessel–nerve cord with the branching of the arteria carotis communis (ACC) into the arteria carotis externa (ACE) and the arteria carotis interna (ACI) was performed. After ligation of ACC and ACE, the ACC was incised and a standardized silicon rubber-coated 6.0 nylon monofilament (Doccol Corporation, Sharon, MA, USA) was inserted. The filament was advanced via the ACI to occlude the origin of the right MCA. After 45 min, the animals were re-anesthetized and the filament was withdrawn to allow reperfusion.

Neurological functions were assessed 24 h and 72 h after surgery by two different scores and mice were sacrificed for the indicated analyses. The experimental design is shown in Appendix A.

### 4.3. Exclusion Criteria

We pre-specified exclusion criteria as follows: (1) death before pre-defined experimental endpoint; (2) drop-out score (overall health condition, spontaneous behavior, weight reduction); and (3) excess of surgery time (duration longer than 10 min in order to exclude a potential influence of prolonged anesthesia and to assure group comparability). The numbers of included/omitted animals in the study of tMCAO with 24 h reperfusion were as follows: 12/2 wt, 12/2 ERK2^wt^, 13/1 RKIP^wt^, and 13/0 RKIP^S153A^. The numbers of included/omitted mice in the study of tMCAO with 72 h reperfusion were as follows: 12/4 wt, 12/6 ERK2^wt^, 10/1 RKIP^wt^, and 10/2 RKIP^S153A^.

### 4.4. Assessment of Functional Outcome

Neurological deficits were characterized according to Bederson [51]. Therefore, animals were put on a surface and their spontaneous movements were recorded and graded as the following: 0, no deficit; 1, forelimb flexion; 2, forelimb flexion plus reduced resistance to lateral push; 3, unidirectional circling; 4, longitudinal spinning; and 5, no movement [52].

To analyze coordination and motor function, the grip test was performed. Animals were put in the middle of a string between two supports and scored as follows: 0, falls off; 1, hangs on to string by one or both fore paws; 2, as for 1, and attempts to climb on to string; 3, hangs on to string by one or both fore paws plus one or both hind paws; 4, hangs on to string by fore and hind paws plus tail wrapped around string; and 5, escape to the supports [52,53].

An assessment of neurological functional outcome was performed 24 h and 72 h after tMCAO.

### 4.5. Determination of Infarct Size

Animals were sacrificed 24 h or 72 h after surgery and brain tissue was removed. The tissue was cut in three coronal sections (2 mm thick) with a mouse brain slice matrix (Harvard Apparatus, Holliston, MA, USA) and slices were stained with 2% 2,3,5-triphenyltetrazolium chloride (TTC; Sigma-Aldrich, Hamburg, Germany; 20 min, 37 °C) in phosphate buffer to visualize the infarct [50].

The calculation of indirect, i.e., corrected for brain edema, infarct volumes was done by volumetry (ImageJ software 1.52a) as follows:V_indirect_ (mm^3^) = V_infarct_ × (1-(V1 − VC)/VC),
whereas the term (V1 − VC) describes the volume difference between the ischemic hemisphere and the control hemisphere and (V1 − VC)/VC represents this difference as a percentage of the control hemisphere [54].

### 4.6. Determination of Blood–Brain Barrier Leakage

To evaluate the permeability of the cerebral vasculature, we injected 2% Evans Blue (Sigma-Aldrich, Hamburg, Germany) tracer diluted in 0.9% NaCl (Fresenius Kabi, Bad Homburg, Germany; i.v.) 1 h after the induction of tMCAO. After 24 h, the animals were euthanized and brain tissue was removed. The tissue was cut in coronal sections (2 mm thick) that were incubated in 4% paraformaldehyde (Sigma-Aldrich, Hamburg, Germany) in the dark for 30 min and then divided into ipsi- and contralateral areas. Brain sections were weighed and incubated with formamide (Sigma-Aldrich, Hamburg, Germany) at 50 °C in the dark for the following 24 h. On the next day, samples were centrifuged for 20 min and 50 µL of the supernatant was collected and transferred into a 96-well plate. The concentration of Evans Blue in the tissue was determined in duplicates by measuring the fluorescence intensity in a microplate reader (INFINITE 200 Pro, TECAN, Männedorf, Switzerland; excitation 620 nm, emission 680 nm) and calculated after linear regression analysis from a standard curve [50].

### 4.7. Protein Extraction and Immunoblot Analysis

We dissected the basal ganglia of contralateral hemispheres from the mouse brains and homogenized the tissue in ice-cold lysis buffer containing 50 mM Tris (Roth, Karlsruhe, Germany; pH 7.4), 300 mM NaCl (Sigma-Aldrich, Hamburg, Germany), 5 mM EDTA (AppliChem, Darmstadt, Germany), 50 mM NaF (Sigma-Aldrich, Hamburg, Germany), 5 mM Na_4_P_2_O_7_ (Sigma-Aldrich, Hamburg, Germany), 0.1 mM Na_3_VO_4_ (Sigma-Aldrich, Hamburg, Germany), 1% (*v*/*v*) Triton-X-100 (AppliChem, Darmstadt, Germany), 1.5 mM NaN_3_ (AppliChem, Darmstadt, Germany), 20 µg/mL soybean trypsin inhibitor (Sigma-Aldrich, Hamburg, Germany), 0.4 mM benzamidine (AppliChem, Darmstadt, Germany), and 1 mM PMSF (AppliChem, Darmstadt, Germany) [55]. After centrifugation at 0 °C and 14,000 rpm for 5 min, supernatants were collected and protein extent was measured using a bicinchoninic acid (BCA) protein assay according to the manufacturer’s instruction (Pierce™ BCA Protein Assay Kit, Thermo Fisher Scientific, Schwerte, Germany). Protein concentration was set to 1 µg/mL and samples were treated with SDS-PAGE loading buffer (50 mM Tris (pH 6.8), 2% (m/V) SDS (AppliChem, Darmstadt, Germany), 10% (m/V) glycerol 85% (Sigma-Aldrich, Hamburg, Germany), 0.0125% (m/V) bromophenol blue (AppliChem, Darmstadt, Germany), 5% (*v*/*v*) β-mercaptoethanol (Roth, Karlsruhe, Germany)). SDS-PAGE was applied for protein separation and the transfer to PVDF membranes was done by wet transfer [20]. Membrane blocking was performed depending on the primary antibody in bovine serum albumin (BSA)-buffer (5% (m/V) BSA (AppliChem, Darmstadt, Germany), 50 mM Tris (pH = 7.4), 150 mM NaCl, 0.2% (*v*/*v*) NP-40 (Sigma-Aldrich, Hamburg, Germany)) or in blocking milk (5% (m/V) non-fat dried milk powder (AppliChem, Darmstadt, Germany), 100 mM NaCl, 10 mM Tris (pH 7.6), 0.1% (*v*/*v*) Tween 20 (AppliChem, Darmstadt, Germany)). Membranes were incubated at 4 °C overnight with the indicated primary antibody at the following dilutions: anti-RKIP (Santa Cruz, Dallas, TX, USA; sc28837) 1:1000, anti-pRKIP (Santa Cruz, Dallas, TX, USA; sc32622) 1:1000, anti-ERK1/2 (cell signaling, Danvers, MA, USA; #9102L) 1:1000, anti-pERK1/2 (cell signaling, Danvers, MA, USA; #9101L) 1:1000, and anti-Gβ (Santa Cruz, Dallas, TX, USA; sc378) 1:5000. After washing, membranes were incubated for 1 h with horseradish peroxidase-conjugated goat anti-rabbit IgG (Sigma-Aldrich, Hamburg, Germany; A0545) 1:10,000. Antibodies were solved in BSA washing buffer (0.25% BSA (m/V), 50 mM Tris (pH 7.6), 150 mM NaCl, 0.2% (*v*/*v*) NP-40). To visualize protein bands, Pierce ECL Plus Western Blotting Substrate (Thermo Fisher Scientific, Schwerte, Germany, 32132) was used. For semiquantitative analysis, immunoblot signals were quantified using ImageJ software 1.52a [47].

### 4.8. RNA Preparation and Quantitative Real-Time PCR

For RNA preparation, brain tissue was first separated into basal ganglia from the ipsilateral hemispheres. The RNeasy Kit (Qiagen, Hilden, Germany) was used for RNA isolation. The Superscript II reverse transcriptase (Thermo Fisher Scientific, Schwerte, Germany) was used for the reverse-transcription of RNA to cDNA [47]. Quantitative real-time polymerase chain reaction (real-time PCR) was performed on the C1000 Thermal Cycler CFX96 (BioRad, Hercules, CA, USA) and relative gene expression levels of endothelin 1 (assay ID: Mm 00438656_m1, Applied Biosystems, Waltham, MA, USA), intercellular adhesion molecule (*ICAM*)-1 (assay ID: Mm 00516023_m1, Applied Biosystems, Waltham, MA, USA), interleukin (*IL*)-*1β* (assay ID: Mm 00434228_m1, Applied Biosystems, Waltham, MA, USA), *IL-6* (assay ID: Mm 00446190_m1, Applied Biosystems, Waltham, MA, USA), tumor necrosis factor-α (*TNF-α*) (assay ID: Mm 00443258_m1, Applied Biosystems, Waltham, MA, USA), and vascular adhesion molecule (*VCAM*)-1 (assay ID: Mm 01320970_m1, Applied Biosystems, Waltham, MA, USA) were analyzed with a fluorescent TaqMan technology. *GAPDH* (TaqMan^®^ Predeveloped Assay Reagent for gene expression, part number: 4352339E, Applied Biosystems, Waltham, MA, USA) was used as an endogenous control. Samples were measured in triplicates and data were analyzed according to the 2^−ΔΔCT^ method [56].

### 4.9. Immunohistochemistry

Cryo-embedded brain sections from mice 24 h after induction of stroke were cut into slices (10 µm thick) and fixed in 5% PFA (Sigma-Aldrich, Hamburg, Germany) in PBS (Sigma-Aldrich, Hamburg, Germany) for the staining of neutrophilic granulocytes. To prevent unspecific binding, the slices were pre-treated with 5% BSA in PBS for 45 min. Invading immune cells were detected by rat anti-mouse Ly-6B.2 alloantigen (neutrophilic granulocytes; MCA771GA, Bio Rad, Hercules, CA, USA; 1:500 in PBS supplemented with 1% BSA, overnight at 4 °C). Then, slides were treated with biotinylated anti-rat IgG (BA-4001, Vector Laboratories, Burlingame, CA, USA; 1:100 in PBS supplemented with 1% BSA; 45 min at room temperature). According to the manufacturer’s instructions (Vectorstain ABC Kit, Peroxidase Standard PK-4000, Vector Laboratories, Burlingame, CA, USA), the coupling to biotinylated peroxidase (POD) was performed. Visualization was performed using chromogen 3,3′-diaminobenzidine (DAB; Kem-En-Tec Diagnostics, Taastrup, Denmark). Neutrophilic granulocytes were then counted at the level of the basal ganglia (0.5 mm anterior from bregma) of four different animals using a Nikon microscope Eclipse 50i. This microscope was equipped with the DS-U3 DS camera control unit and the NIS-Elements software (Nikon, Tokio, Japan) [52].

### 4.10. Apoptosis Measurements

Twenty-four hours after tMCAO, apoptotic neurons in the injured brain were visualized by double immunostaining with NeuN and TUNEL reagent. Cryo-embedded brains were cut in 6 μm thick brain slices and fixed in ice-cold acetone (AppliChem, Darmstadt, Germany) for 10 min. After the blocking of epitopes by incubation of the brain tissue in 5% BSA (AppliChem, Darmstadt, Germany) in PBS containing 1% donkey serum (Dianova, Hamburg, Germany) and 0.3% Triton-X-100 (AppliChem, Darmstadt, Germany) for 1 h, a mouse antibody to NeuN (Merck, Darmstadt, Germany; MAB377, 1:1000 in PBS) was applied overnight at 4 °C. Afterwards, proteins were detected by 1 h of incubation with Cy5 conjugated AffiniPure Donkey Anti-Mouse (Jackson Immuno Research, Cambridgeshire, United Kingdom, 715-175-150) secondary antibody at a dilution of 1:100 in 1% BSA in PBS. The in situ cell death detection kit TMR red was used for the detection of TUNEL positive cells (Roche, Penzberg, Germany, 12 156 792 910) following the manufacturer’s instructions. Samples were analyzed using immunofluorescence microscopy in a 20-fold magnification (Leica, Wetzlar, Germany; DM6B). To determine neuron density, NeuN-positive cells were counted in comparable brain regions from one slice per animal in the ipsi- and contralateral hemisphere. The number of dead neurons was assessed by counting TUNEL/NeuN double positive cells in the same brain sections.

### 4.11. Sample Preparation of Mouse Brain Tissue for Proteomics Analysis

For lysis, punches of snap-frozen mouse brain were prepared and the following buffer was used: 50 mM Tris-HCl, 150 mM NaCl, 1% (m/V) SDS with cOmplete™ ULTRA protease inhibitor (Sigma-Aldrich, Hamburg, Germany), and PhosSTOP (Roche, Penzberg, Germany; pH 7.8). For our experiment, 300 μL of the respective lysis buffer and approximately 20 glass beads were added to the punched tissue (~8 mg). Next, the tissue was disrupted by sonication using 20 cycles consisting of 30 s on/30 s off using a Bioruptor Plus (Diagenode, Denville, NJ, USA) at 4 °C. BCA assay was used according to the manufacturer’s instruction to determine protein concentration (Pierce BCA protein assay kit).

Approximately 100 µg of proteins was reduced in 10 mM dithiothreitol (DTT; AppliChem, Darmstadt, Germany; 30 min, at 56 °C) and alkylated in iodoacetamide (IAA; AppliChem, Darmstadt, Germany); 30 mM, 30 min, room temperature, dark). The remaining IAA was quenched with 30 mM DTT for a further 15 min at room temperature.

Samples were digested using the S-Trap™ (Protifi, Huntington, NY, USA) procedure according to the manufacturer’s instruction. After acidifying the samples by the addition of aqueous phosphoric acid (12%; Sigma-Aldrich, Hamburg, Germany), dilution was accomplished with an S-Trap binding buffer (methanol (90%; Sigma-Aldrich, Hamburg, Germany), triethylammonium bicarbonate (100 mM; Sigma-Aldrich, Hamburg, Germany); pH 7.1). Proteins were loaded onto S-Trap columns including centrifugation steps and filter-based tryptic digestion was accomplished (2 h, 47 °C, trypsin (ratio of 1:20; Promega, Madison, WI, USA)). Thereafter, peptide elution was done by applying eluting steps (10 mM ABC, following 0.1% formic acid (FA; Sigma-Aldrich, Hamburg, Germany), and finally acetonitrile (ACN; Sigma-Aldrich, Hamburg, Germany; 80%)) [57]. Lyophilized peptides were rehydrated in trifluoroacetic acid (TFA; Sigma-Aldrich, Hamburg, Germany; 0.1%) for subsequent quality control [58] and LC-MS/MS analysis.

### 4.12. LC-MS/MS Analysis

A total of 1 μg of the respective peptide samples was separated on an Ultimate 3000 Rapid Separation Liquid chromatography (RSLC) nano system with a ProFlow flow control device coupled to a Q Exactive HF orbitrap mass spectrometer (Thermo Scientific, Schwerte, Germany). For peptide concentration, a trapping column was used (Acclaim C18 PepMap100, 100 μm, 2 cm, Thermo Fisher Scientific, Schwerte, Germany); 0.1% TFA (Sigma-Aldrich, Hamburg, Germany), flowrate 10 μL/min) [59]. For the following reversed phase chromatography (Acclaim C18 PepMap100, 75 μm, 50 cm), we used a binary gradient (solvent A 0.1% FA (Sigma-Aldrich, Hamburg, Germany)/ solvent B 84% ACN (Sigma-Aldrich, Hamburg, Germany) with 0.1% FA; 5% B for 3 min, linear increase to 25% for 102 min, a further linear increase to 33% for 10 min, and then a final linear increase to 95% for 2 min followed by a linear decrease to 5% for 5 min). For MS survey scans, the following settings were used: MS was operated in data dependent acquisition mode (DDA) with full MS scans from 300 to 1600 m/z (resolution 60,000) with the polysiloxane ion at 371.10124 m/z as lock mass [60]. Maximum injection time was set to 120 ms. The automatic gain control (AGC) was set to 1E6. For fragmentation, the 15 most intense ions (above the threshold ion count of 5E3) were chosen at a normalized collision energy (nCE) of 27% in each cycle, following each survey scan. Fragment ions were acquired (resolution 15,000) with an AGC of 5E4 and a maximum injection time of 50 ms. Dynamic exclusion was set to 15 s [57].

### 4.13. Data Analysis

For all data processing, the Proteome Discoverer software 2.3.0.523 (Thermo Scientific, Schwerte, Germany) was used and searches were done in a target/decoy mode against a mouse UniProt database (downloaded 30 November 2019, UniProt (www.uniprot.org)) using the MASCOT algorithm. The following search parameters were used: precursor and fragment ion tolerances of 10 ppm and 0.02 Da for MS and MS/MS; a trypsin set as the enzyme with a maximum of two missed cleavages; carbamidomethylation of a cysteines set as the fixed modification and the oxidation of methionines was set as a dynamic modification; and using a Percolator false discovery rate set to 0.01 for both peptide and protein identifications [61]. A label-free quantification (LFQ) analysis was performed including replicates (number stated in corresponding tables) for each condition. Proteins were considered as significantly regulated with *p* ≥ 0.05 after identification with at least two unique peptides and a log2 ratio ≥ 1 (2-fold enrichment).

The raw data of the mass spectrometry analyses are available under ProteomeXchange Consortium via the PRIDE [62] partner repository with the dataset identifier PXD030067 and 10.6019/PXD030067.

### 4.14. Statistics

For statistical analyses, the GraphPad Prism 7 software package (San Diego, CA, USA) was used. Results of the Bederson score and grip test are expressed as ordinal values. All other results are given as mean ± standard error (mean ± SEM). If only two groups were compared, we used an unpaired, two-tailed Student´s *t*-test analysis. For more than two groups, one-way ANOVA with post hoc Tukey correction was applied. The results of the Bederson score and grip test were compared by a Kruskal–Wallis test. *p* < 0.05 was considered statistically significant.

## Figures and Tables

**Figure 1 ijms-23-00706-f001:**
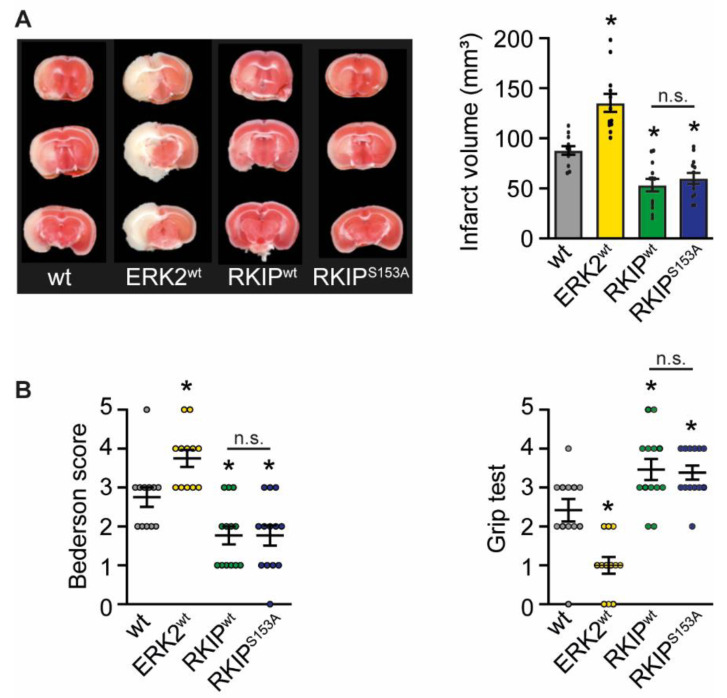
Overexpression of ERK2^wt^ increases infarct size and neurological deficits whereas RKIP^wt^ and RKIP^S153A^ show opposite effects 24 h after induction of stroke. Characterization of wild-type (wt) mice (*n* = 12), mice with ubiquitous overexpression of ERK2^wt^ (ERK2^wt^; *n* = 12), RKIP^wt^ (RKIP^wt^; *n* = 13), or a phosphorylation-deficient mutant of RKIP^S153A^ (RKIP^S153A^; *n* = 13) 24 h after induction of stroke by transient middle cerebral artery occlusion. (**A**) Analysis of infarct volume by 2,3,5-triphenyltetrazolium chloride (TTC) staining. Left: Representative TTC stained coronal brain sections. Right: Quantification of infarct volume. (**B**) Analysis of neurological functions using Bederson score and grip test. Error bars are mean ± SEM; *n* represents the number of animals. * *p* < 0.05 versus wt. n.s.—Not significant.

**Figure 2 ijms-23-00706-f002:**
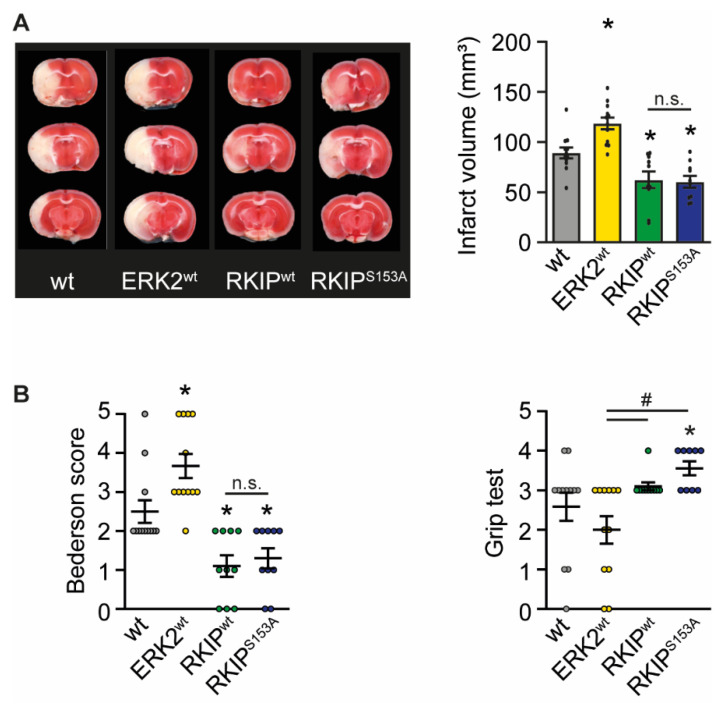
Overexpression of ERK2^wt^ increases infarct size and neurological deficits whereas RKIP^wt^ and RKIP^S153A^ show opposite effects three days after tMCAO. Characterization of wild-type (wt) mice, mice with ubiquitous overexpression of ERK2^wt^ (ERK2^wt^), RKIP^wt^ (RKIP^wt^), or a phosphorylation-deficient mutant of RKIP^S153A^ (RKIP^S153A^) three days after induction of stroke by transient middle cerebral artery occlusion. (**A**) Analysis of infarct volume by 2,3,5-triphenyltetrazolium chloride (TTC) staining. Left: Representative TTC stained coronal brain sections. Right: Quantification of infarct volume (wt, ERK2^wt^: *n* = 12; RKIP^wt^, RKIP^S153A^: *n* = 10). (**B**) Analysis of neurological functions using Bederson score (wt, ERK2^wt^: *n* = 12; RKIP^wt^, RKIP^S153A^: *n* = 10) and grip test (wt, ERK2^wt^: *n* = 12; RKIP^wt^: *n* = 10; RKIP^S153A^: *n* = 9). Error bars are mean ± SEM; *n* represents the number of animals. * *p* < 0.05 versus wt. # *p* < 0.05 versus the indicated group. n.s.—Not significant.

**Figure 3 ijms-23-00706-f003:**
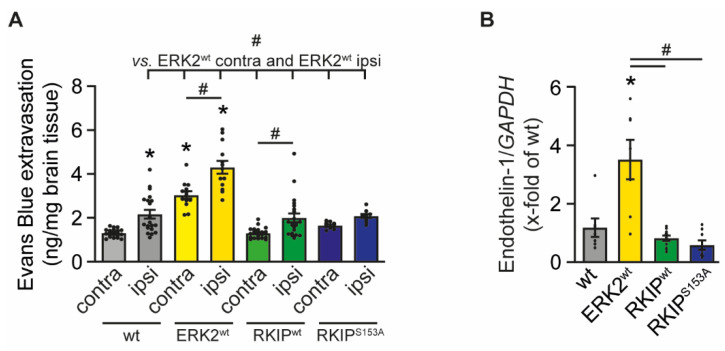
ERK2^wt^ overexpression reduces blood–brain barrier stability. Analysis of the stability of blood–brain barrier in wild-type mice (wt), mice with ubiquitous overexpression of ERK2^wt^ (ERK2^wt^), RKIP^wt^ (RKIP^wt^), or a phosphorylation-deficient mutant of RKIP^S153A^ (RKIP^S153A^) 24 h after induction of stroke by transient middle cerebral artery occlusion. (**A**) Quantitative analysis of the extravasation of the vascular tracer Evans Blue in the contralateral (contra) and ipsilateral (ipsi) hemisphere (wt: *n* = 20; ERK2^wt^: *n* = 13; RKIP^wt^: *n* = 22; RKIP^S153A^: *n* = 9). (**B**) mRNA expression of endothelin-1 in the ipsilateral basal ganglia normalized to glycerinaldehyde-3-phosphate dehydrogenase (*GAPDH*) (wt, ERK2^wt^: *n* = 7; RKIP^wt^: *n* = 11; RKIP^S153A^: *n* = 8). Error bars are mean ± SEM; *n* represents the number of animals. * *p* < 0.05 versus wt. # *p* < 0.05 versus the indicated group.

**Figure 4 ijms-23-00706-f004:**
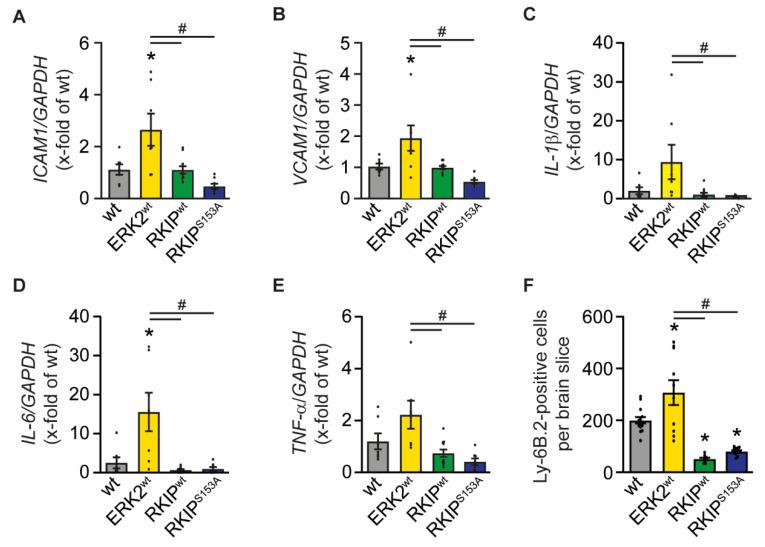
Overexpression of ERK2^wt^ increases inflammatory signaling after stroke whereas overexpression of RKIP and RKIP^S153A^ reduce it. Analysis of inflammation in wild-type mice (wt), mice with ubiquitous overexpression of ERK2^wt^ (ERK2^wt^), RKIP^wt^ (RKIP^wt^), or a phosphorylation-deficient mutant of RKIP^S153A^ (RKIP^S153A^) 24 h after induction of stroke by transient middle cerebral artery occlusion. (**A**–**E**) mRNA expression of intercellular adhesion molecule 1 (*ICAM1*, **A**), vascular adhesion molecule 1 (*VCAM1*, **B**), interleukin-1β (*IL-1β*, **C**), interleukin-6 (*IL-6*, **D**), and tumor necrosis factor-α (*TNF-α*, **E**) in the ipsilateral basal ganglia normalized to glycerinaldehyde-3-phosphate dehydrogenase (*GAPDH*) (wt, ERK2^wt^: *n* = 7; RKIP^wt^: *n* = 11; RKIP^S153A^: *n* = 8). (**F**) Quantitative analysis of immunostaining against Ly-6B.2-positive neutrophils at the level of the basal ganglia (*n* = 4 animals per group with 14 analyzed pictures for wt, 10 for ERK2^wt^, 8 for RKIP^wt^, and 13 for RKIP^S153A^). Error bars are mean ± SEM; *n* represents the number of animals. * *p* < 0.05 versus wt. # *p* < 0.05 versus the indicated group.

**Figure 5 ijms-23-00706-f005:**
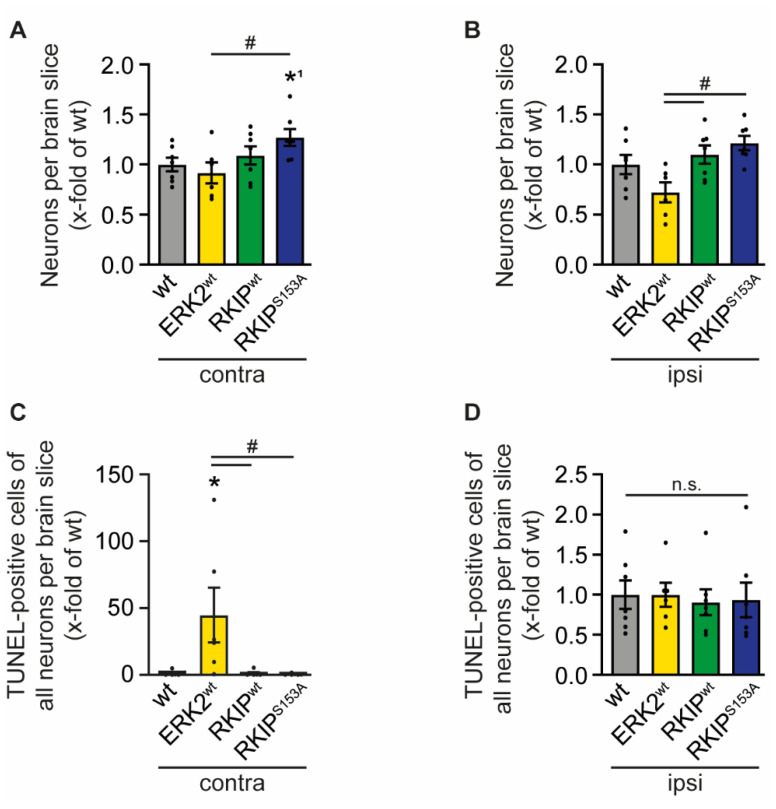
ERK2^wt^ overexpression promotes neuronal death. Wild-type mice (wt), mice with ubiquitous overexpression of ERK2^wt^ (ERK2^wt^), RKIP^wt^ (RKIP^wt^), or a phosphorylation-deficient mutant of RKIP^S153A^ (RKIP^S153A^) were subjected to 45 min of transient middle cerebral artery occlusion (tMCAO). Twenty-four hours after induction of stroke by tMCAO, brain sections were incubated with NeuN to stain neurons in the contralateral (contra; wt, RKIP^wt^, RKIP^S153A^: *n* = 7; ERK2^wt^: *n* = 6) (**A**) and ipsilateral (ipsi; wt, RKIP^wt^, RKIP^S153A^: *n* = 7; ERK2^wt^: *n* = 6) (**B**) hemisphere and co-stained with TUNEL-reagent to analyze apoptotic neurons in the contralateral (contra; *n* = 6) (**C**) and ipsilateral (ipsi; wt, RKIP^wt^, RKIP^S153A^: *n* = 7; ERK2^wt^: *n* = 6) (**D**) hemisphere. Error bars are mean ± SEM; *n* represents the number of animals. * *p* < 0.05 versus wt in an ANOVA test. *^1^
*p* < 0.05 versus wt in a *t*-test. # *p* < 0.05 versus the indicated group. n.s.—Not significant.

**Figure 6 ijms-23-00706-f006:**
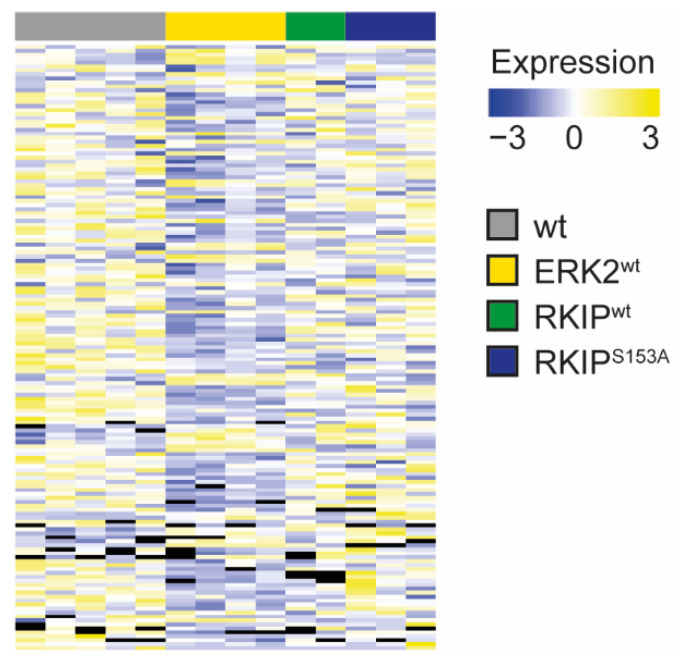
ERK2^wt^ overexpression induces the biggest changes compared to wild-type controls, whereas RKIP overexpression was similar to wild-type. Mass spectrometric analysis of proteins in the ipsilateral basal ganglia of wild-type mice (wt, *n* = 5), mice with ubiquitous overexpression of ERK2^wt^ (ERK2^wt^, *n* = 4), RKIP^wt^ (RKIP^wt^, *n* = 2), or a phosphorylation-deficient mutant of RKIP^S153A^ (RKIP^S153A^, *n* = 3) 24 h after induction of stroke by transient middle cerebral artery occlusion. Heat map visualization of the significantly regulated proteins between wt and ERK2^wt^. Black bars indicate missing data. For protein list and individual values, refer to Appendix A.

## Data Availability

Raw data of mass spectrometry analyses are available under ProteomeXchange Consortium via the PRIDE [62] partner repository with the dataset identifier PXD030067 and 10.6019/PXD030067. Raw data for additional information required to interpret, replicate, or build on the findings of this study are available from the corresponding author upon reasonable request.

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
