# Peer review of "ERK1/2 Activity Is Critical for the Outcome of Ischemic Stroke"

_ijms, 2022, doi:10.3390/ijms23020706_

Round 1

Reviewer 1 Report

the study presented by Schanbacher et al, was conducted very elegantly and according to good practice. The article is very well organized and easily read. 
I still have some discussion points that I would suggest. 
concerning the statistical analysis, I suggest for the scores to analyze with non-parametric tests.
For the future, when you have multidimensional data, it will be necessary to apply an adapted methodology like the supervised principal component analysis. 

For the discussion: I think that despite the strength of your model, we cannot yet rule out a beneficial role of transient activation of the ERK pathway. Moreover, it may be that animals with a permanently invalidated ERK pathway have a predisposition to suffer more due to the lack of expression of a more resistant profile (see your proteomic data). 

Author Response

We thank the reviewer for the positive response, the interest in our work and the critical suggestions.

With regards to the statistical analysis of the scores of the neurological functions, we have now applied the Kruskal-Wallis-test. We thank the reviewer for this comment and will consider it also in future.

With regards to the discussion, we have now included an additional comment to further clarify the limitation/restriction of the model used (line 277-279).

Reviewer 2 Report

The Authors focused on a study of the ERK1/2 activity is critical for the outcome of ischemic stroke. This is an interesting and comprehensive study. The article is well structured. Six figures in the text are very clearly written.

In my opinion:

  • The abstract presents an accurate description of this study.
  • An Authors was conducted adequate literature review.
  • The references support the rationale for reporting the study.
  • The subjects are described adequately.
  • The management of the study is effectively described.
  • Valid and reliable outcome measures are utilized.
  • The conclusions are appropriate.

Overall impression about the quality of the study is good.

Key points to consider:

Line 19 – explain the abbreviation wt - first time in article

Line 35 – “2,3,4” please fix it to “2-4”

Line 45 – explain the abbreviation miR-1 - first time in article

Line 49 – “15,16,17,18” please fix it to “15-18”

Line 54 – explain the abbreviation CAG - first time in article

Line 70 – “…their wild-type (wt) littermates …” please fix it to “…their wt littermates …”

Line 92 – explain the abbreviation I/R - first time in article

Line 108 – “…of the blood brain barrier (BBB) is one …” please fix it to “…of the BBB is one …”

Line 231 – “33,34,35” please fix it to “33-35”

Line 231 – “36,37,38” please fix it to “36-38”

Line 262 – “15,16,17,18” please fix it to “15-18”

Please explain the abbreviations in whole paper - if first time in article.

Author Response

We are grateful for the overall positive comments and the helpful suggestions.

We adjusted all the mentioned key points and those changes are marked in the corrected manuscript. We also followed the suggestion to check the explanation of all abbreviations.